# Technology-Supported University Courses for Increasing University Students’ Physical Activity Levels: A Systematic Review and Set of Design Principles for Future Practice

**DOI:** 10.3390/ijerph18115947

**Published:** 2021-06-01

**Authors:** Kuston Sultoni, Louisa Peralta, Wayne Cotton

**Affiliations:** 1Sydney School of Education and Social Work, University of Sydney, Sydney, NSW 2006, Australia; louisa.peralta@sydney.edu.au (L.P.); wayne.cotton@sydney.edu.au (W.C.); 2Faculty of Sports and Health Education, Universitas Pendidikan Indonesia, Bandung 40154, Jawa Barat, Indonesia

**Keywords:** college, course, design principles, physical activity, technology, university

## Abstract

Physical activity levels tend to decrease as adolescents’ transition to adulthood. University course-based interventions utilising technology are a promising idea to combat this decrease. This review aims to systematically identify, critically appraise, and summarise the best available evidence regarding technology-supported university courses that aim to increase student’s physical activity levels. The second aim is to create initial design principles that will inform future practice in the area. Data Sources: CINAHL, ERIC, MEDLINE, ProQuest, PsycINFO, Scopus, SPORTDiscus, Web of Science. Search dates from January 2010 to December 2020. Study Inclusion: RCT or non-RCT or quasi-experimental studies describing university course-based interventions using technology that aim to increase the physical activity levels of university students. Data Extraction: Source (country), methods, participants, interventions, theoretical frameworks and type of technologies, outcome and measurement instrument, and results. Data Synthesis: Systematic review. Result: A total of 1939 articles were identified through databases. Six studies met the inclusion criteria. Conclusion: Four of the six included studies reported significant increases in university students’ physical activity levels. An analysis of the six included studies identified four design principles that future course designers could utilise as they develop technology-supported university courses that aim to increase the physical activity levels of university students. Further work is required to test the effectiveness of these four design principles.

## 1. Introduction

Promoting physical activity among university students is beneficial not only for students’ physical health and wellbeing but also for mental health and wellbeing, stress tolerance, and academic performance [1,2,3]. Despite the benefits, physical activity levels tend to decrease in this age group [4,5,6,7,8]. Studies have shown that most university students do not meet physical activity guidelines [9,10]. These results suggest that promoting physical activity among university students is essential.

Studies targeting university students’ physical activity levels have gained momentum over the previous five years [11,12,13,14]. Belogianni and Baldwin conducted an overview study of the type of interventions targeting diet, physical activity, and weight-related outcomes among university students [11]. Their findings show that there were four main groups of intervention (environmental, face-to-face, electronic, and combined interventions). The subtype of face-to-face course-based interventions had a moderate effect on physical activity behaviour. In addition, physical activity outcomes were the most frequently reported, compared with other health-related outcomes. Plotnikoff and colleagues’ systematic review of physical activity interventions [14] found that 19 of 29 studies were successful at significantly increasing university students’ physical activity levels. However, this review only focused on the effectiveness of the interventions and did not extract and compare the intervention delivery strategies that led to these increases in physical activity. Another systematic review of interventions promoting physical activity among university students was conducted by Maselli and colleagues [13]. This review identified and discussed methodologies, effective strategies, and deficiencies in the interventions designed to promote university students’ physical activity levels. The findings suggest that an effective intervention should address the multifaceted aspects of human behaviour such as motives, knowledge, perceived or necessary skills to perform actions, and self-regulation [13]. A more recent systematic review conducted by García-Álvarez and Faubel [12] focused on the specific strategies of effective interventions and measurement tools for assessing university students’ physical activity levels. The findings reported that 70% of the included studies found a significant increase in a physical activity outcome, with multifaceted courses being the most prevalent strategy used (9 of 13 studies). However, these systematic reviews have not focused on measuring the impact of university course-based physical activity interventions implemented in a university setting.

Several strategies have been reported to improve the quality of physical activity courses delivered in the university setting [15,16,17,18]. One strategy is that universities should have organisational systems to manage learning materials, teacher management and development, and technology to improve learning processes [15,16,17,18]. Presently, the use of technology for pedagogical and teaching practices varies among physical activity course instructors at universities. The most used technological platform is the learning management system [19]. However, the research emphasises that the teacher should provide a creative technology-integrated learning experience while achieving a teaching goal [20,21]. Deciding what kind of technology to be used as a pedagogical tool becomes a crucial issue to accommodate teachers’ knowledge and attitudes toward using technology and the technologies available for use [22]. However, it is not yet clear which technologies and features are the most effective for increasing university students’ physical activity levels through a university course-based setting.

Therefore, this systematic review aims to identify, critically appraise, and summarise the available evidence regarding the effectiveness of technology-supported university courses for increasing university students’ physical activity levels. This systematic review focuses on the effectiveness of a technology-supported university-course intervention targeting university students’ physical activity levels and the types and features of the technologies used. The study’s secondary aims are to create a set of initial design principles to advise future research on how to develop effective technology-supported university courses that aim to increase the physical activity levels of university students. Design principles can be defined as a series of prescriptive theoretical understandings [23] best expressed in active terms [24], which take the form of heuristic statements [25] that can be used to inform practice or to guide the design of interventions [26]. The initial design principles that emerge from this review can inform and prescribe the practice of physical activity promotion through university courses that utilise technology.

## 2. Materials and Methods

This systematic review was undertaken following the guideline of the Preferred Reporting Items for Systematic Reviews and Meta-Analyses (PRISMA) [27]. The protocol of the study was registered with PROSPERO (CRD42020210327).

### 2.1. Data Sources

The sources of electronic bibliographic databases include CINAHL (EBSCO Information Services, USA), ERIC (The Institute of Education Sciences of the U.S. Department of Education, USA), MEDLINE (National Library of Medicine, USA), ProQuest (Cambridge Information Group, USA), PsycINFO (American Psychological Association, USA), Scopus (Elsevier, The Netherlands), SPORTDiscus (EBSCO Information Services, USA), and Web of Science (Clarivate Analytics, USA).Search dates from 1 January 2010 to 31 December 2020. Only manuscripts written in English were included in the review, and additional records identified through other sources were also considered.

### 2.2. Search Strategy

The search strategy included terms referring to 4 main concepts: (1) population (“Higher Education” OR “University students” OR College*); (2) intervention: course-based intervention using technology (technolog* OR online* OR “learning management system” OR LMS OR website* OR “wearable devices” OR App OR Apps OR “mobile application*” OR “smartphone application*” OR Android OR “activity tracker” OR GPS OR “Global Positioning System” OR “blended learning” OR web-based OR “web based” OR virtual OR internet*); (3) type of study RCT/non-RCT/quasi-experiment (test OR RCT OR randomi* OR control OR trial OR evaluat* OR quasi-exper* OR cluster OR intervention*); and (4) outcome (“Physical activ*” OR “Physically active” OR “Physical education”). The complete search strategies for all databases are provided in Appendix A.

### 2.3. Study Selection and Data Extraction

The first author (KS) searched the stated databases with consultation with the university librarian. The retrieved studies from all databases were imported into the ENDNOTE X9 reference manager. Then, duplicate studies and non-English articles were removed, as well as posters, proceedings, pilot studies, study protocols, and review articles. The first author (KS) screened the titles and abstracts of selected studies to exclude non-eligible studies. Then, two authors (KS and LP) reviewed the full-text articles and made inclusion decisions based on the inclusion criteria. There was an almost perfect level of agreement (97.4%, Cohen’s Kappa = 0.907). However, when discrepancies occurred, the third author (WC) also reviewed the full-text and resolved any disagreements through discussion. Once the included studies were decided, the following data were extracted into Microsoft Excel: source (country), methods, participants, intervention, theoretical framework and type of technology, outcome and measurement instrument, and results.

### 2.4. Methodological Quality Assessment

The methodological quality assessments for the selected studies were assessed using a 10-item quality assessment scale derived from Van Sluijs and colleagues [28] (See Table 1). Two authors (LP and WC) independently reviewed each study to determine whether the studies met (Yes) or did not meet (No) the criteria. Then, the number of “Yes” studies was accumulated and defined as a high-quality methodological study if a RCT study scored six or higher or a non-RCT study scored five or higher [28].

### 2.5. Strategy for Data Synthesis

A meta-analysis approach was not feasible due to the included studies’ conceptual heterogeneity, such as the interventions, outcomes, and measurements. Therefore, a narrative synthesis approach was conducted [29]. The design principles were generated using a thematic analysis rubric adapted from Mansfield and colleague’s study [30].

## 3. Results

A total of 1939 articles were identified through electronic bibliographic databases searching (CINALH: 27, Eric: 74, MEDLINE: 212, ProQuest: 470, PsycINFO: 88, Scopus: 473, SPORTDiscus: 76, and Web of Science: 519). After removing 899 duplicates, 1040 titles and abstracts were screened. After the screening of the titles and abstracts, 38 papers were selected for full-text review. In the full-text review, 32 studies were excluded because they did not meet the inclusion criteria. Finally, six studies met the inclusion criteria for the current review [31,32,33,34,35,36]. The process is summarised and presented in the PRISMA flow diagram (Figure 1).

### 3.1. Characteristics of the Included Studies

As shown in Table 2, four studies were conducted in the USA [32,33,34,36] and one each in China [31] and Japan [35], with a total population of 642 university students with a mean age of 17 to 24 years. Three studies were randomised control trials [31,33,35], and three were nonrandomised controlled trials [32,34,36]. Two studies used a combination of two theoretical frameworks to underpin the intervention. One study used Bandura’s social–cognitive theory (SCT) and the health belief model (HBM) [35], and one used a combination of SCT and the transtheoretical model [TTM] [33]. The other four studies did not specify a theoretical framework. Three studies had physical activity and nutrition behaviour as multiple outcomes [31,32,36], while three other studies had physical activity as a single outcome [33,34,35]. Regarding the instruments for measuring physical activity, four studies used self-report measurements. Two studies used the International Physical Activity Questionnaire (IPAQ) [31,35]; one study used a questionnaire that was developed based on the American College of Sports Medicine training guideline for exercise testing and prescription [36], and the other study used physical activity diaries through a mobile application (app) based on IPAQ’s concept [32]. While two studies used objective measurement for physical activity, one study used a right hip pedometer Yamax Digi-Walker [34] and one study used an ActiGraph accelerometer [33].

### 3.2. Summary of Interventions

The technology-supported courses varied among the studies. The courses of two studies specifically targeted physical activity as a single facet [33,35], while the other four studies targeted physical activity as part of a multifaceted approach [31,32,34,36]. The course duration of three studies were one semester [32,34,36]. One study was four months in length [35], and one study was 15 weeks [33]. The other study was only 21 days in duration [31]. Three studies did not mention who taught the course [32,35,36]. One study mentioned that the course was taught by a trained master’s degree graduate instructor [33]. The course in one study was taught by the researcher [34]. The other study mentioned that the course was taught by the program leader with help from operators, health assistants, dietitians, and a sports coach [31]. Regarding the course content, two studies did not report the content details [32,36]. Four studies reported the course content in the paper, which was the core curriculum or traditional course content [31,33,34,35]. The details of the course’s content can be seen in Table 3.

In regard to the type of technology utilised in the included studies, two studies used an online website (internet-based physical activity program [35] and online wellness course [36]). Two studies used an activity tracker as an intervention (Misfit Flash [33] and Fitbit Zip [34]), and two studies used smartphones (social media [31] and a developed mobile application (app) [32]). The features of the activity trackers that were utilised in the two studies [33,34] were similar and included tracking activity such as walking (total steps), running, and calories burned. The online website in Okazaki’s study [35], besides providing course content, also had features of goal-setting, physical activity scheduling, and web-based quizzes. Meanwhile in Everhart and Dimon’s study [36], the online website only provided multimedia for the course content. In Wang’s study [31], the course material was provided through social media. This platform also had features such as comments, reminders, and voice broadcasts. Whilst in Krzyzanowski’s study [32], their Rams Have Heart mobile application had a feature for diarising dietary and physical activity.

### 3.3. Summary of Intervention Effects on Physical Activity Outcomes

The two studies that used an online website [35,36] reported a positive effect on physical activity. Okazaki and colleagues [35] developed the internet-based physical activity program (i-PAP), which was underpinned by SCT and HBM, and therefore included a website containing goal-setting, scheduling, self-monitoring, physical activity information, quizzes, and energy-expenditure calculations. Participants in the intervention group had access to the website and received advice according to the physical activity reported, while the control group had no treatment. Participants who at baseline did not engage in regular university sports activities in the intervention group showed a significant increase in energy expenditure after the intervention [F (2, 72) = 3.5, *p* < 0.05] when compared with the control group. Everhart and Dimon’s study [36] compared three courses (online, traditional, and combined wellness courses). Participants who completed the blended delivery format (i.e., the combined wellness course) or the traditional classroom format increased their weekly cardiovascular exercise engagement significantly more than those who completed the web-based format (*p* < 0.041).

Of the two studies that used an activity tracker [33,34], only one of the studies reported an effect on physical activity [34]. Rote’s study [34] compared three groups of participants (Intervention I: health course + Fitbit group, Intervention II: health course only, and Control: humanities course only). The mean score for steps/day of the intervention I groups significantly increased (*p* = 0.014). The mean score for steps/day for intervention II and control groups did not significantly change (*p* = 0.621, *p* = 0.132). In the study that implemented an activity tracker that had no effect on physical activity levels [33], participants used an activity tracker to measure physical activity daily. The objectively measured MVPA minutes in the intervention group were not changed in the hypothesised direction but tended to remain stable over time.

Of the two studies using a smartphone [31,32], one study significantly increased university students’ physical activity levels [31]. Wang and colleagues [31] evaluated the 21-day social media WeChat program, which contained dietary advice, exercise encouragement, and healthy-habits reminders as a supplement to an eight-week Application of Nutrition and Health Care course. Students’ physical activity levels in the intervention group were enhanced; 48 participants were at a low physical activity level at baseline, and 26 of them moved to a higher level after 21 days of intervention (*p* = 0.004). While, in Krzyzanowski and colleague’s [32] study, although the Rams Have Heart App integrated self-reported health screening with health education, diary tracking, and user feedback modules, activity levels in both minutes and METS decreased over time.

Two of the six included studies were underpinned by a combination of theoretical frameworks [33,35]. Of those two studies, only one had a positive effect on university student’s physical activity levels at the completion of the intervention [35]. Okazaki and colleagues [35] used several theoretical framework-informed strategies such as goal setting, self-monitoring, and energy-expenditure calculations for physical activity. However, in the study that had no effect on physical activity levels, Kim and colleagues’ [33] used a larger number of theoretical framework-informed strategies such as personalized goal-setting, activity-tracking, calorie calculations, behavioural feedback and social support. Perhaps the timing and frequency of the implementation of these strategies is a consideration not yet determined. In addition, it should be noted that a number of other studies in this review [31,32,34] implemented several of these strategies but did not detail whether a theoretical framework was used to justify these strategies.

### 3.4. Quality Assessment of Included Studies

The results of the methodological quality assessment can be viewed in Table 4. Two studies met > five assessment criteria [33,35]. One study met four criteria [34]; one study met two criteria [32]; one study only met one criterion [31], and one study did not meet any criteria [36]. Five of the six studies (83%) presented the key baseline characteristic separately for the treatment group [31,32,33,34,35]. Three studies (50%) reported their findings using validated measures [33,34,35]. Only two studies (33%) reported blinded outcome variable assessments [32,35] and summary results for each group [33,34]. Only Kim’s study [33] clearly described their randomisation procedure, intention to treat analysis for the outcome(s) and accounted for potential confounders for outcome analysis. Only the Okazaki study [35] reported a drop out rate and assessed the outcomes a minimum of 6 months after pretest. Only the Rote study [34] reported a power calculation, with the study adequately powered to detect hypothesised relationships.

### 3.5. Design Principles

Four design principles were generated based on a synthesis of the included studies:

Design principle one: Technology-supported university courses for increasing students’ physical activity levels should provide physical activity knowledge with extensive feedback and interaction between teachers and students.

Technology-supported university courses should provide physical activity knowledge to increase student’s physical activity levels and extensive feedback on their knowledge. First and foremost, transferring knowledge is considered the vital element of a course. Hence, if physical activity is the course’s target, then physical activity knowledge should be prioritised. Three references identified by this review provide empirical evidence to support the design principle [31,35,36]. In Okazaki et al.’s study [35]. a website was utilised to provide the course material containing physical activity knowledge and a web-based quiz to track student knowledge. While in Wang et al. [31], physical activity knowledge was delivered in two-way communication processes with “well-designed topics” of the health education program (P.340). In contrast, Everhart and Dimon’s [36] study designed a web-based course that did not embed instructor feedback. The intervention did not affect students’ physical activity behaviours.

Design principle two: Technology-supported university courses for increasing student’s physical activity levels should utilise a simple and familiar technological device.

Empirical support for this design principle is provided by all studies included in this review. The example of simple and familiar devices is shown in four studies that utilise an online website, smartphone application or social media [31,34,35,36]. This kind of technology was used in the studies because it was considered a novel and widespread trend among university students. While, in the study that utilised an activity tracker, both studies reported participants having a technical issue [33,34], and only one of these studies [34] had a positive effect on physical activity levels.

Design principle three: Technology-supported university courses for increasing student’s physical activity levels should allow students to create individual goal-setting, track their achievement, and receive personalised feedback from the teacher.

Empirical evidence to support this design principle is provided by two studies that reported implementing strategies that allowed participants to set goals, track achievement, and receive personal feedback from the instructor about their physical activity levels. Subsequently, intervention participants’ were able to increase their physical activity levels at follow-up [34,35]. One of the studies included in this review detailed the theoretical frameworks that underpinned the study design and were able to implement a number of theoretical framework-informed strategies throughout the intervention to increase students’ physical activity levels [35]. In addition to feedback from the instructor, participants in Wang and colleagues’ study [31] were also under intense supervision from the course instructor (health assistants), and the result showed that students’ physical activity levels increased.

Design principle four: Technology-supported university courses for increasing student’s physical activity levels should provide exercise examples that guide students to do more physical activity in their own time, independently and outside of the dedicated course time.

Two references identified by this review provide empirical evidence to support this design principle [31,33]. In Wang and colleagues’ [31] study, an example guide was provided by a hyperlink from an app called Keep. This app was used to guide a 5 min exercise challenge as a daily task. In the Kim and colleagues’ [33] study, “student-led practice sessions were provided to implement their own individualised physical activity and fitness plans, which were developed through a series of course assignments related to self-regulatory of behavioural change techniques” (P. 1895).

## 4. Discussion

The purpose of the current study was to review the technology-supported university courses designed to increase university students’ physical activity levels and the types and features of the technology used. The findings show the diverse types of technology implemented in course-based interventions targeting physical activity in a university setting. Three types of technology utilised in the selected studies were online websites, activity trackers, and smartphones (social media and mobile app). The findings also revealed that four of the studies attained four or fewer of the criteria on the assessment tool, demonstrating a high risk of bias, therefore restricting the ability to draw firm conclusions of the effectiveness of these six studies.

Four of the six included studies reported significant increases in university students’ physical activity levels. Two studies utilised an online website integrated within the course [35,36] and provided course materials, assignments, and quizzes on the website. The online websites also had additional features related to behavioural change theory, such as setting goals, schedules, and other physical activity promotion features. The results from these two studies are in line with previous studies that utilised online websites in non-courses-based university settings [37,38,39,40]. For example, the IMPACT study by Cholewa and Irwin [37] showed that online record-keeping helps students become more active as they are more efficient in terms of monitoring their own physical activity. In addition, another study conducted by Magoc and colleagues [38] used a web-based physical activity intervention with materials structured based on the social cognitive theory. The study’s findings showed that the intervention had the intended effect on physical activity. A different study on a web non-course-based intervention to increase student physical activity was carried out by Duan and colleagues [39]. The results showed a significant improvement in students’ physical activity levels; however, the dropout rate was high. They argued this occurred due to it being a non-course-based intervention, as well as a lack of intervention features, such as appropriate website layout and formatting. This result is consistent with a previous study by Skar and colleagues [40]. The delivery of the internet site with material was deemed attractive and engaging by university students, and therefore, the intervention was successful at increasing students’ physical activity levels. Hence, using an internet website to deliver a physical activity intervention is a promising strategy due to practicability, however the material needs to be well-prepared, presented appropriately, and involve user engagement.

One study [34] in the current review that showed significant increases in university students’ physical activity levels utilised a Fitbit activity tracker that has features, such as tracking activity, walking (total step), running, and calories burned. This is supported by other studies focusing on increasing university students’ physical activity levels. For example Melton and colleagues study [41] examined the feasibility and acceptability of the Jawbone Up accelerometer in a non-course-based setting. Results emphasised that positive feedback from participants was high, demonstrating the feasibility and acceptability of using accelerometers with university students. In spite of the advantages of using wearable devices, there are issues around the negative motivational consequences [42], demonstrating peer pressure [43], and the affordance of the device [44]. Therefore, future course-based interventions should consider using activity trackers combined with other technologies, such as websites or mobile apps.

One study [34] in this review reported an increase in students’ physical activity levels through social media. This platform had valuable features such as comments, reminders and voice broadcasts to enhance teacher and student communication. This finding is contrary to previous studies that utilised social media to increase students’ physical activity levels in non-course-based university settings [45,46]. This research shows that the combination of web-based education and social networking (Facebook) had no effect on students’ physical activity levels and perceptions [45], and social networking mobile apps connected with a wearable tracker also had no effect [46]. Although social media has promising features and potential reach, further understanding how this component can be used to support the delivery of a physical activity intervention should be pursued.

Four of the six included studies did not specify a theoretical framework that informed the design of the intervention courses. This is contrary to a previous review conducted by Maselli, Ward, Gobbi and Carraro [13] that reported only three of 28 included studies did not specify a theoretical framework. Although most of the studies included in Maselli and colleagues’ [13] review were non-course-based and therefore reported theoretical frameworks to define outcomes and content, it is still unclear which of the intervention components and theoretical-framework-informed strategies were the most effective in increasing physical activity. This is due to a lack of qualitative or process data, limited information on how the theoretical frameworks informed the intervention designs, and the different ways theoretical framework-informed strategies were implemented across studies [13]. For future course-based interventions there is a strong need for these to be underpinned by at least one theoretical framework and to outline how theoretical-framework-informed strategies are implemented to determine the extent to which these have an effect on physical activity outcomes at post-intervention and at follow-up. This is an important consideration for ensuring that course-based interventions are sustainable.

Regarding data collection instruments, the present review identified that four of the six included studies (67%) measured physical activity levels using a self-report questionnaire [31,32,35,36]. This is consistent with a previous review conducted by García-Álvarez and Faubel [12] that reported that 70% of included studies utilised validated questionnaires for measuring physical activity. While self-reported questionnaires are valid, more objective measures are recommended in future studies [47].

Of the two studies that specifically targeted physical activity as a single facet [33,35], only one study (50%) had a significant effect on physical activity outcomes [35]. In addition, three of the four studies (75%) that implemented a multifaceted approach had a significant impact on physical activity outcomes [31,34,36]. These findings are consistent with one other review conducted by Plotnikoff and colleagues [14], which found seven of the 11 (64%) studies targeted physical activity, and 11 of 18 (61%) studies that targeted multiple behaviours had significant effects on the physical activity levels of university students. However, considering the high risk of bias in two studies in this review [31,36], multifaceted interventions require further consideration in future research.

The current review also led to the creation of four initial design principles to advise future development in the area. However, caution should be used when using these four design principles, as they were constructed using evidence from only six included studies in this review, which displayed medium to low methodological quality. Further needs analysis research is required to implement the initial design principles. This will intensify the design principles, particularly if collaboration with practitioners are enacted, as this is crucial in defining implementation issues [48,49]. Then, these initial design principles should be tested and evaluated in controlled trials.

The present systematic review was undertaken following the PRISMA guidelines to ensure the quality of the report. Nevertheless, this study has several limitations. First, although we searched the articles from eight different databases and used a broad search strategy, the eligible study number was low. A possible explanation for this might be that we only selected articles written in English. Second, we did not conduct a meta-analysis approach due to selected studies’ conceptual heterogeneity, such as the intervention approaches, outcomes, and measurements. Hence, the strength of evidence presents a single summary estimate on the effect on technology-supported university courses for increasing university student’s physical activity levels were not obtained.

## 5. Conclusions

This study identified six studies which utilised three types of technologies that were embedded in university course-based interventions that targeted student’s physical activity levels. The technologies included online websites, activity trackers, and smartphone applications. Four of the six included studies reported significant increases in university students’ physical activity levels. Four design principles were generated from the six included studies that future course designers could utilise to inform the development of technology-supported university courses that aim to increase the physical activity levels of university students. Further work is required to implement, test, refine, and evaluate the design principles.

## Figures and Tables

**Figure 1 ijerph-18-05947-f001:**
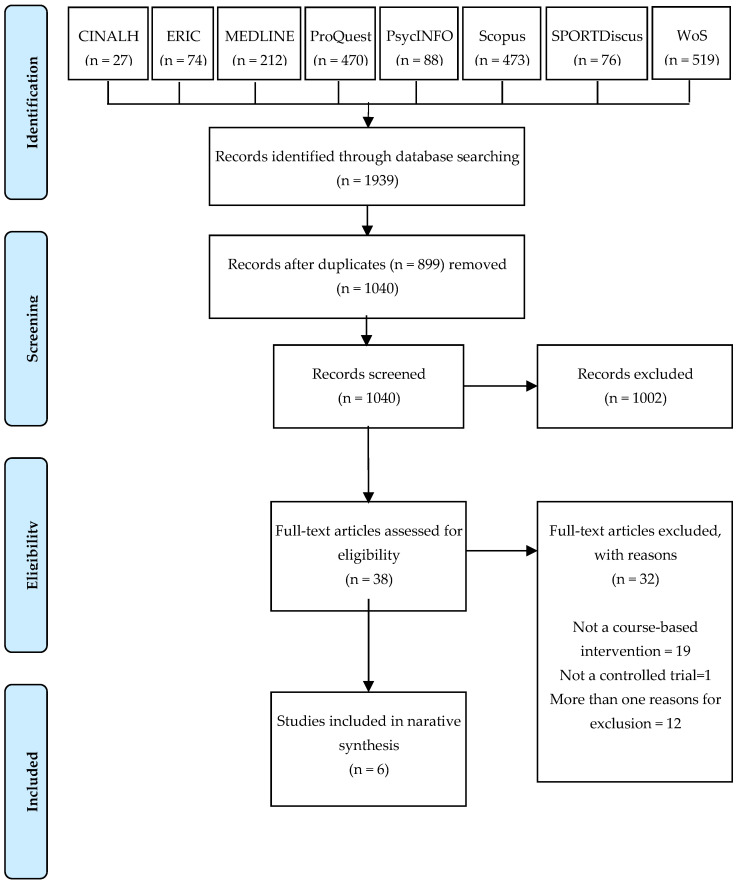
PRISMA flow diagram for the search and inclusion for identification of articles.

**Table 1 ijerph-18-05947-t001:** Methodological quality assessment criteria.

Criterion	Description
A	Key baseline characteristics are presented separately for treatment groups (age and one relevant outcome) and for randomised controlled trials, positive if baseline outcomes were statistically tested and results of tests were provided.
B	Randomisation procedure clearly and explicitly described and adequately carried out in randomised controlled trials (generation of allocation sequence, allocation concealment and implementation)
C	Validated measures of outcomes assessed (validation in same age group reported and/or cited)
D	Drop out reported and ≤20% for <6-month follow-up or ≤30% for ≥6-month follow-up
E	Blinded outcome variable assessments
F	Outcomes assessed a minimum of 6 months after pretest.
G	Intention to treat analysis for outcome(s) (participants analysed in the group they were originally allocated to, and participants not excluded from analyses because of noncompliance to treatment or because of some missing data)
H	Potential confounders accounted for in outcome analysis (e.g., baseline score, group/cluster, age)
I	Summary results for each group + treatment effect (difference between groups) + its precision (e.g., 95% confidence interval)
J	Power calculation reported, and the study was adequately powered to detect hypothesised relationships

**Table 2 ijerph-18-05947-t002:** Descriptive data.

Source (Country)	Methods	Participants	Interventions	Theoretical Frameworks & Type of Technologies	Outcomes & Measurement Instruments	Results
Wang et al., 2020 [31] (China)	Nonrandomised control intervention.	N = 110 (Age 18 ± 1); Intervention = 87 (M = 37; F = 50); control = 23 (M = 8; F = 15);	INTERVENTION: Participants took a Nutrition and Health Care course for eight weeks and downloaded the WeChat app. Participants also received 21 days of dietary advice, exercise encouragement, healthy habits reminders during the program starting from week 5 of the offline course. CONTROL: Participants only took a Nutrition and Health Care course for eight weeks and downloaded the WeChat app.	TF: Non-specify Social media (WeChat)	Eating Habits: dietary intake estimates Physical Activity: International physical activity questionnaire (IPAQ) Physical Fitness: the per-minute times of push-ups, time of quiet squat down against the wall, and time of plank	Physical Activity (PA) in intervention group were enhanced, 48 participants were at low PA level in baseline, and 26 of them moved to higher level after 21 days intervention (*p* = 0.004) Daily food intake (vegetable, fruit, milk, and dairy products) of intervention group were significantly improved (all *p* < 0.05).
Krzyzanowski et al., 2020 [32] (USA)	Nonrandomised control intervention	N = 109 (Age from 17 to 24 years); Intervention = 55; Control = 54	INTERVENTION: Participants took the CVD risk-reduction course and were directed to use the Rams Have Heart app. The app integrates self-reported health screening with health education, diary tracking, and user feedback modules to acquire data and assess progress. CONTROL: Participants only took the traditional health course and did not use the App	TF: Non-specify Rams Have Heart (Mobile App)	Healthy behaviour: Developed App Fruit and vegetable consumption: Developed App Physical Activity: Metabolic Equivalent (METs) generated from developed App based on IPAQ Functionality, usability, and adherence of the developed App	Activity levels in both minutes and METS decreased over time.Fruit and vegetable intake trended slightly upward as the study progressed. 49% students in cohort 2 and 45% in cohort 3 used the Rams Have Heart app at least once. Over the course of the fall semester, app participation dropped off gradually until exam week when most students no longer participated.
Kim et al., 2018 [33] (USA)	Cluster randomised control trial	N = 187 (Age 20.32 ± 1.57); Intervention = 101 (M = 34; F = 67); control = 86 (M = 37; F = 49);	INTERVENTION: Participants took a Physical Activity Instructional Program (PAIP) course and wore a Misfit Flash (activity tracker that can be worn with a clasp or watch band) for 15 weeks. There was no intervention component mandated in the curriculum of the PAIP course. Participant were encouraged to track their activity level and use all of the features of the Misfit App on a daily basis. CONTROL: Participants only took the standardised core curriculum of PAIP.	TF: Social Cognitive Theory and the Transtheoretical Model Activity Tracker and App (Misfit Flash)	Physical Activity: ActiGraph Actitrainer (ActiGraph LLC, Pensacola, FL, USA)	The objectively measured MVPA minutes in the intervention group were not changed in the hypothesised direction but tended to remain stable over time. Sedentary time and LPA minutes in the intervention group were significantly increased and decreased over time, respectively.
Rote, 2017 [34] (USA)	Controlled trial study	N = 56; Intervention1 = 24 age 20.4 ± 2.5 (M = 8; F = 16); Intervention2 = 14 age 22.8 + 9.1 (M = 6; F = 8); control = 18 age 19.1 ± 1.6 (M = 12; F = 6);	INTERVENTION 1: Participants took an experimental version of a health course (education + Fitbit group). INTERVENTION 2: Participants took a traditional version of a health course (education-only group), CONTROL: Participants only took a humanities course.	TF: Non-specify Activity tracker and App (Fitbit Zip)	Physical activity: right hip Pedometer Yamax Digi-Walker Participants’ experience with Fitbit: open-ended questions	The mean score for steps/day of the Education + Fitbit group significantly increased (*p* = 0.014) The mean score for steps/day within the Education only group and control group did not significantly change (*p* = 0.621, *p* = 0.132)
Okazaki et al., 2014 [35] (Japan)	Randomised controlled trial	N = 77; Intervention = 49 age 19.1 ± 1.3 (M = 35; F = 14); control = 28 age 19.4 ± 1.2 (M = 15; F = 13);	INTERVENTION: Participants took an internet-based physical activity program (i-PAP) and had access to a website containing goal-setting, scheduling, self-monitoring, PA information (health behaviour skills, body images, training), quizzes, and energy-expenditure calculations. Participants received advice according to PA reported. CONTROL: Participants had no treatment.	TF: social cognitive theory, health belief model, online website	Physical activity: International Physical Activity Questionnaire (IPAQ) and the stages of change scale for physical activity (SOC)	Only the DS subgroup (did not engage in university sport) in the intervention group exhibited significant increases in energy expenditures compared with the control group [F (2, 72) = 3.5, *p* < 0.05]. The energy expenditure of the DS subgroup in the intervention group was significantly higher than in the control group at the 8-month follow-up [F (1, 36) = 4.3, *p* < 0.05].
Everhart and Dimon, 2013 [36] (USA)	Three group pre- and post-test	N = 103 Intervention 1 = 27 Intervention 2 = 20 Control = 56	INTERVENTION 1: Participants took an online web-based wellness course delivery. INTERVENTION 2: Participants took a combined online and traditional wellness course. CONTROL: Participants took a traditional wellness course.	TF: unspecified web-based instructional delivery	Physical Activity: developed questionnaire Nutritional eating habit: developed questionnaire	The blended group improved cardiovascular exercise duration habits more than the web-based group, and the regular classroom group increased their minutes per week of cardiovascular workouts significantly more than students completing the course totally online (*p* < 0.041).

**Table 3 ijerph-18-05947-t003:** An overview of the university courses incorporated in the included studies.

Source (Country)	Course Name	Duration and Instructors	Program/Content	Technology Features	Design Principles Informed
Wang et al., 2020 [31] (China)	Application of Nutrition and Health Care	DURATION: Eight-week course (21 days intervention) INSTRUCTOR: Program leader, operator, health assistance, dietitian, sport coach	21-day health relating topics: Latest nutritional concept; Healthy lifestyle; How to choose carbohydrate; How to choose protein; How to select vegetables and fruits; Dealing with stress; Weight management; The practical use of fat; The importance of mineral substances; The importance of vitamins; Diet management in college students; The importance of water; Balanced diet in college students; Nutrition in daily life; Dealing with academic pressure; How to choose snacks; How to choose take-out food; Health promoting methods; How to arrange three meals for a day; Self-control of diet; How to control weight after weight loss	Social media (WeChat) Share, Post, Comment, Log, Reminder, Voice Broadcast, and Publish popular science article of physical activity and nutrition materials.	1, 2, 3, 4
Krzyzanowski et al., 2020 [32] (USA)	Cardiovascular disease (CVD) course	DURARTION: 1 Semester (Cohort 2 was under study from fall 2017 to spring 2018, and cohort 3,from fall 2018 to spring 2019) INSTRUCTOR: not specified	The fruit and vegetable intake module The physical activity module	Mobile App (Rams Have Heart) Tracking diaries, health information, personal feedback, and support modules—with an illustrative icon, a function label, and a brief descriptive statement to aid understanding.	2
Kim et al., 2018 [33] (USA)	A physical activity instructional program (PAIP)	DURARTION: 15 weeks. INSTRUCTOR: trained master’s degree graduate instructors	Basic exercise (e.g., principles of physical fitness development, health benefits of PA) Behavioural sciences (e.g., healthy habits, goal setting)	Activity Tracker and App (Misfit Flash) Activity tracking (total step, calories burned, distance ran and/or walked, and activity points calculated from three measures), personalised goal setting, standardised behavioural feedback, and social community support.	2, 3, 4
Rote, 2017 [34] (USA)	Introductory health course	DURATION: during the spring semester (January to May) of 2014 INSTRUCTOR: The researchers served as instructors.	Nutrition Physical activity Weight management Mental Emotional health	Activity Tracker and App (Fitbit Zip) Tracks step taken, distance travelled, and calories burned per day, steps/day, dietary intake and sleep habits	2
Okazaki et al., 2014 [35] (Japan)	Internet-based physical activity program (i-PAP)	DURATION: 4 months INSTRUCTOR: not specified	Physical activity and exercise Quantity and quality of physical activity and exercise Promotion and maintenance of physical activity and exercise Weight control Receipts and disbursements balance of energy and nutrition Calculation of consumption energy Four type of training contractions: isotonic, eccentric, isometric, isokinetic Strength-training methods, strength-training meals-plan, stretch-training methods, sport injury summary	Online Website goal-setting, scheduling, web-based quizzes, PDF reader	1, 3
Everhart and Dimon, 2013 [36] (USA)	Wellness course	DURATION: Fall 2009 semester INSTRUCTOR: not specified	Nutrition and The benefits of physical activity	**Online Website** provides a variety of multimedia and other web-based resources on the course websites	1, 2

**Table 4 ijerph-18-05947-t004:** Methodological quality assessment.

Study	Methodological Quality Assessment Items										Criteria Met (n)
	A	B	C	D	E	F	G	H	I	J	
Wang et al., 2020 [31]	yes	no	no	no	no	no	no	no	no	no	1
Krzyzanowski et al., 2020 [32]	yes	no	no	no	yes	no	no	no	no	no	2
Kim et al., 2018 [33]	yes	yes	yes	no	no	no	yes	yes	yes	no	6
Rote, 2017 [34]	yes	no	yes	no	no	no	no	no	yes	yes	4
Okazaki et al., 2014 [35]	yes	no	yes	yes	yes	yes	no	no	no	no	5
Everhart and Dimon, 2013 [36]	no	no	no	no	no	no	no	no	no	no	0

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
