# Peer review of "Technology-Supported University Courses for Increasing University Students’ Physical Activity Levels: A Systematic Review and Set of Design Principles for Future Practice"

_ijerph, 2021, doi:10.3390/ijerph18115947_

Round 1

Reviewer 1 Report

Currently, with the pandemic of COVID-19, students' daily life and learning behavior are severely restricted. In such a situation, from the standpoint of a university faculty member who is in charge of sports practice like us, how to increase the physical activity levels of students is a field of great interest. Current systematic review covered the most recent reports, and exactly address to this point. Although the authors reviewed only six articles, but the four course design principles proposed based on this systematic review were convincing. Therefore, we believe that this review provides implications for all faculty members designing health education courses at universities. I would like to comment only two points on the Discussion section.

Students undergo course-based intervention at least partly for credits, so it is natural that the adherence is high. It is very doubtful that students will continue to work after the course, and I don't think there are any reports mentioning it. In fact, the authors also argue in the 2nd paragraph of Discussion that there are many dropouts in non-courses-based university settings, citing previous reports. The authors emphasize the importance of making webpages practical and attractive as a countermeasure for dropouts. Is there any other incentive that enhances student participation in non-courses-based university settings? In the case of course-based intervention, is there any way for students to continue working after the course is over? Further discussion is needed on this point.

In the 3rd paragraph of Discussion, while the authors mentioned the potential use of activity trackers, they also described that the use of these devices involves technical issues and can also have a negative impact on students' motivation for physical activity. It is clear from the results of many previous studies that the use of activity trackers, not limited to students, will continue to play an important role in efforts to increase the physical activity levels of people. How can we overcome the shortcomings of activity trackers, for example by using a combination with an intervention using online website? This is a part of the reader's interest and requires further discussion.

Reviewer 2 Report

Congratulations to the authors. In my opinion the article is well thought out. The objectives are well defined. The methodological process is rigorous.  And the results and conclusions provide valuable information in relation to an interesting subject of study.

I have detected only two small errors. 
Error line 123 and 124
Line 277 278 there is a full stop at the beginning of the sentence.

Reviewer 3 Report

Dear authors,

Thank you for the opportunity to read your systematic review. It is a well-written manuscript that focuses on an important topic, not only in the US but also in European countries. I only have some minor comments regarding content. However, there is a lot of work regarding formatting. Therefore, I would like to encourage the authors to carefully format the manuscript because there are a lot of flaws (e.g., problems with reference in line 123, changes in font, and Table 3 is included twice).

Did the authors calculate kappa for the second selection step? This is an important aspect for the quality of a systematic review, therefore, Cohen's kappa should be included.

Round 2

Reviewer 3 Report

I would like to thank the authors for revising the manuscript according the reviewers' comments.